# Assessment of Paraoxonase 1 and Arylesterase Activities and Lipid Profile in Bodybuilders: A Comparative Study of Physical Activity and Anthropometry on Atherosclerosis

**DOI:** 10.3390/medicina60101717

**Published:** 2024-10-20

**Authors:** Hakim Celik, Mehmed Zahid Tuysuz, Yakup Aktas, Mehmet Ali Eren, Recep Demirbag

**Affiliations:** 1Department of Physiology, Faculty of Medicine, Harran University, Sanliurfa 63290, Turkey; 2Department of Biophysics, Faculty of Medicine, Harran University, Sanliurfa 63290, Turkey; mz.tuysuz@harran.edu.tr; 3Department Coaching Education, School of Physical Education and Sports, Harran University, Sanliurfa 63290, Turkey; yakupaktas@harran.edu.tr; 4Department of Endocrinology and Metabolism, Faculty of Medicine, Harran University, Sanliurfa 63290, Turkey; mehmetali.eren@lokmanhekim.edu.tr; 5Department of Cardiology, Faculty of Medicine, Harran University, Sanliurfa 63290, Turkey; rdemirbag@harran.edu.tr

**Keywords:** bodybuilding, atherosclerosis, physical activity, antioxidant enzymes, lipid profiles, paraoxonase 1, arylesterase

## Abstract

*Background and Objectives*: Atherosclerosis, driven by dyslipidaemia and oxidative stress, is a leading cause of cardiovascular morbidity and mortality. This study evaluates the effects of vigorous-intensity bodybuilding exercise (VIBBE) on atherosclerosis biomarkers—including paraoxonase-1 (PON1) and arylesterase (ARE) activities—and lipid profiles in male bodybuilders who do not use anabolic-androgenic steroids. Comparisons were made with individuals engaged in moderate-intensity aerobic exercise (MIAE), as well as overweight/obese sedentary (OOS) and normal-weight sedentary (NWS) individuals. *Materials and Methods*: A cross-sectional study was conducted involving 122 healthy male participants aged 18–45 years, divided into four groups: VIBBE (*n* = 31), OOS (*n* = 30), MIAE (*n* = 32), and NWS (*n* = 29). Anthropometric assessments were performed, and fasting blood samples were collected for biochemical analyses, including lipid profiles and PON1 and ARE activities. Statistical analyses compared the groups and evaluated correlations between adiposity measures and atherosclerosis biomarkers. *Results*: The VIBBE group exhibited significantly lower levels of low-density lipoprotein cholesterol (LDL-C), triglycerides (TG), and logarithm of the TG to high-density lipoprotein cholesterol (HDL-C) ratio [log(TG/HDL-C)] compared to the OOS group (*p* < 0.05 for all), indicating improved lipid profiles. However, these improvements were not significant when compared to the NWS group (*p* > 0.05), suggesting that VIBBE may not provide additional lipid profile benefits beyond those associated with normal weight status. PON1 and ARE activities were significantly lower in the VIBBE group compared to the MIAE group (*p* < 0.05 for both), suggesting that VIBBE may not effectively enhance antioxidant defences. Correlation analyses revealed significant inverse relationships between PON1 and ARE activities and adiposity measures, including body mass index (BMI), waist-to-hip ratio (WHR), waist-to-height ratio (WHtR), body fat percentage (BFP), fat mass index (FMI), and obesity degree (OD) (*p* < 0.05 for all). Positive correlations were observed between oxLDL and log(TG/HDL-C) and adiposity measures (*p* < 0.05 for all). *Conclusions*: Vigorous-intensity bodybuilding exercise improves certain lipid parameters compared to sedentary obese individuals but does not significantly enhance antioxidant enzyme activities or further improve lipid profiles beyond those observed in normal-weight sedentary men. Conversely, moderate-intensity aerobic exercise significantly enhances PON1 and ARE activities and improves lipid profiles, offering superior cardiovascular benefits. These findings underscore the importance of incorporating moderate-intensity aerobic exercise into physical activity guidelines to optimize cardiovascular health by balancing improvements in lipid metabolism with enhanced antioxidant defences.

## 1. Introduction

Atherosclerosis is a chronic, progressive disease characterised by the accumulation of lipids, fibrous elements, and other substances within the arterial walls. It remains a leading cause of cardiovascular diseases (CVDs) worldwide, contributing to increased morbidity and mortality rates [1,2,3]. This progressive condition results in the formation of atheromatous plaques, leading to arterial occlusion and subsequent cardiovascular events such as myocardial infarction and stroke [1]. The pathogenesis of atherosclerosis is complex and multifactorial, involving genetic factors, environmental influences, and lifestyle behaviours [4].

Central to the development and progression of atherosclerosis are dyslipidaemia and oxidative stress [5]. Dyslipidaemia is characterised by elevated levels of low-density lipoprotein cholesterol (LDL-C) and triglycerides (TG), accompanied by reduced levels of high-density lipoprotein cholesterol (HDL-C). This imbalance forms an atherogenic lipid profile that promotes plaque formation [5,6]. In addition, oxidative stress, often signalled by increased oxidised low-density lipoprotein (oxLDL) levels, exacerbates endothelial dysfunction and inflammation, accelerating the development of atherosclerotic plaques [5,7].

In recent years, significant progress has been made in understanding the roles of oxidative stress-related enzymes, particularly paraoxonase 1 (PON1) and arylesterase (ARE), in cardiovascular disease. These enzymes, which are closely linked to HDL-C, play a vital role in protecting against oxidative modifications of LDL-C, a key event in the development of atherosclerosis [8,9]. The antioxidant activity of PON1 and ARE helps mitigate oxidative stress, a major factor in the initiation and progression of atherosclerosis [9]. Additionally, emerging evidence shows that the functions of PON1 and ARE extend beyond lipid metabolism. For instance, their activities are influenced by different intensities of physical activity, highlighting their potential role in exercise-induced adaptations that improve cardiovascular health [10].

The effects of physical activity on PON1 and ARE activities have garnered increasing attention. Vigorous-intensity bodybuilding, a form of resistance training focused on muscle hypertrophy and strength, has been recognised for its potential cardiovascular benefits [11]. Unlike aerobic exercises that primarily enhance cardiovascular endurance [12], bodybuilding emphasises muscle mass and strength through high-intensity resistance training [11,13]. This type of training improves insulin sensitivity and glucose metabolism [14], and can positively influence lipid profiles by increasing HDL-C and reducing LDL-C and TG levels [15]. However, the specific effects of vigorous-intensity bodybuilding on atherosclerosis biomarkers such as the TG to HDL-C ratio [log(TG/HDL-C)], oxLDL levels, and PON1 and ARE activities remain underexplored, necessitating further research.

Moderate-intensity aerobic exercise, often recommended for general health, has well-established effects on improving lipid metabolism and reducing oxidative stress [16,17,18]. Activities such as walking, jogging, and light aerobic exercises are widely prescribed for cardiovascular health [19]. Moderate-intensity exercise has been well-documented to increase HDL-C levels while lowering LDL-C and TG [20]. Additionally, it is associated with enhanced PON1 and ARE activities, which help protect lipoproteins from oxidative modification [21]. Although the benefits of moderate-intensity aerobic exercise are well-documented, comparative studies investigating how it differs from vigorous-intensity bodybuilding in terms of influencing atherosclerosis biomarkers are limited, highlighting the need for further investigation.

Recent findings have expanded the role of PON1 and ARE beyond cardiovascular health, linking these enzymes to a variety of systemic conditions. Reduced PON1 and ARE activities have been associated with neurodegenerative diseases, such as Alzheimer’s and Parkinson’s [22], as well as metabolic disorders like type 2 diabetes, where they exacerbate oxidative stress and inflammation [23]. Their diminished activity is also implicated in non-alcoholic fatty liver disease (NAFLD) [24], chronic kidney disease (CKD) [25], and autoimmune conditions [9], further highlighting their significance in modulating systemic oxidative damage. Given these associations, interventions aimed at enhancing PON1 and ARE activities, such as targeted exercise regimens, could offer therapeutic potential in managing these diverse conditions, reinforcing the importance of these enzymes in broader health contexts.

Obesity, characterised by excessive adiposity, significantly increases the risk for atherosclerosis and other cardiovascular diseases [26]. Obesity aggravates dyslipidaemia, promotes systemic inflammation, and elevates oxidative stress, all of which contribute to the progression of atherosclerosis [27,28]. The chronic inflammatory state associated with obesity results in the release of pro-inflammatory cytokines and adipokines, further promoting endothelial dysfunction and lipid abnormalities [29]. The absence of physical activity in sedentary individuals exacerbates these risks, emphasising the importance of exercise in mitigating the adverse effects of obesity [30].

This study aims to systematically evaluate the impact of vigorous-intensity bodybuilding exercises on atherosclerosis biomarkers, including PON1 and ARE activities, oxLDL, LDL-C, HDL-C, TG levels, and the log(TG/HDL-C) ratio. By comparing these biomarkers among individuals engaged in vigorous-intensity bodybuilding, those participating in moderate-intensity aerobic exercise, and sedentary obese individuals, this study seeks to elucidate the potential cardioprotective effects of bodybuilding. The novelty of this study lies in its examination of these biomarkers in bodybuilders who do not use anabolic-androgenic steroids, contributing to the expanding body of research on exercise physiology and cardiovascular health.

## 2. Materials and Methods

### 2.1. Participants

This study involved 122 healthy male volunteers aged between 18 and 45 years, all of whom were free from chronic illnesses. Each participant underwent routine health examinations to confirm their eligibility and provided written informed consent. The research protocol was approved by the Harran University Clinical Research Ethics Committee on 5 September 2022 (Approval number: HRÜ/22.17.18) and was conducted in full compliance with the principles of the Declaration of Helsinki. To gather detailed information, participants completed a comprehensive questionnaire that assessed their weekly exercise workload, dietary habits, and other routine activities, as detailed in Table 1. The exclusion criteria were carefully defined to exclude individuals with a recent history of injury or illness, or those using medications that could influence atherosclerosis biomarkers. Volunteers meeting the inclusion criteria were assigned to the study in four distinct groups as outlined below, based on the type of physical activity they engaged in and their body mass index (BMI).

Vigorous-Intensity Bodybuilding Exercise (VIBBE) Group: This cohort comprised 31 volunteers with a BMI exceeding 25, all of whom had engaged in regular anaerobic weightlifting exercises for a minimum of three years. Their exercise regimen consisted of an average of five days per week, with each session lasting an average of 81.29 min of vigorous-intensity training. This group represents individuals committed to high-intensity, resistance-based physical activity, aiming to explore the effects of such strenuous exercise on various health markers.

Overweight/Obese Sedentary (OOS) Group: This group included 30 overweight or obese participants with a BMI comparable to those in the VIBBE group. These individuals had maintained a sedentary lifestyle, refraining from participation in any structured exercise programs for at least three years. This group’s data serves to contrast the health outcomes of sedentary behaviour against those of active lifestyles within similar body mass index (BMI) ranges.

Moderate-Intensity Aerobic Exercise (MIAE) Group: This group consisted of 32 participants with a BMI under 25 who had been consistently performing moderate-intensity aerobic exercises, such as brisk walking or light jogging, for at least three years. Their routine averaged three days per week, with each session lasting an average of 74.06 min. This group represents individuals engaged in regular aerobic activity of moderate intensity, allowing for the examination of its impact on health parameters in comparison to more vigorous or sedentary lifestyles.

Normal-Weight Sedentary (NWS) Group: This group involved 29 normal-weight sedentary participants with a BMI under 25, similar to the MIAE group. These individuals had not engaged in any structured exercise programs for at least three years. The inclusion of this group facilitates a comparative analysis of the health outcomes between sedentary and physically active individuals within the normal-weight range, providing a comprehensive understanding of the effects of exercise on health across different BMI categories.

### 2.2. Anthropometric Assessments

Anthropometric and body composition measurements were meticulously recorded by an experienced researcher to ensure accuracy and reliability. Heights were measured using a stadiometer (Holtain Ltd., Crymych, UK) with a precision of ±1 mm. Body composition was assessed using bioelectrical impedance analysis (BIA) with the InBody 720 device (Biospace Co., Seoul, Republic of Korea). Participants adhered to stringent pre-test guidelines, which included avoiding physical activity for 24 h, fasting for 12 h, and refraining from alcohol or tobacco consumption to minimize measurement variability. On the measurement day, participants wore light clothing and ensured their bladders were empty to avoid any confounding variables. BMI was calculated as weight in kilograms divided by the square of height in meters (kg/m^2^). The body surface area (BSA) was computed using the Mosteller formula: BSA (m^2^) = √([Height(cm) × Weight(kg)]/3600) [31]. Fat-free mass (FFM) was determined by subtracting body fat weight from total body weight, using body fat percentage obtained from BIA. The fat-free mass index (FFMI) was calculated by dividing fat-free mass FFM by height squared (kg/m^2^), thus standardizing body composition measures across different heights. Similarly, the fat mass index (FMI) was determined by dividing fat mass (kg) by height squared (m^2^), providing a standardized measure of body fat relative to height [32]. Skinfold thickness measurements were conducted at three specific anatomical sites—abdominal, suprailiac, and triceps—using a skinfold calliper (Holtain Ltd., Crymych, UK). These measurements followed the standardized procedures as outlined by the international society for the advancement of kinanthropometry (ISAK). Each measurement was taken three times, and the average value was recorded to enhance reliability. Table 1 comprehensively details the collected skinfold data along with other anthropometric measurements and lipid profiles.

### 2.3. Assessment of Physical Activity

Physical activity levels in the VIBBE and MIAE groups were assessed using the short form of the international physical activity questionnaire (SF-IPAQ), which gathers data on the number of days and duration of participants’ activities over the preceding week [33]. The reported activities were quantified by assigning a metabolic equivalent of task (MET) value to each type of activity [34]. The estimated energy expenditure due to physical activity, expressed as MET-min/week, was calculated by multiplying the frequency, intensity, and duration of each reported activity [35]. Participants in the OOS and NWS groups, who did not report any physical activity, were classified as sedentary.

### 2.4. Biochemical Analyses

Venous blood samples (5 mL) were collected from each participant following at least 8 h of overnight fasting and after the completion of anthropometric measurements. These blood samples were then centrifuged at 1500× *g* for 10 min within 2 h of collection. The separated serum samples were subsequently stored at −86 °C until further analysis.

PON1 and ARE activities were measured using commercial kits (Relassay, Gaziantep, Turkey) [36]. PON1 activity was determined by monitoring the increase in absorbance at 412 nm at 37 °C due to the hydrolysis of paraoxon (diethyl-p-nitrophenyl phosphate) to p-nitrophenol. The amount of p-nitrophenol produced was calculated using the molar extinction coefficient at pH 8.5 (18,290 M^−1^cm^−1^). ARE activity was measured using phenyl acetate as the substrate, with the produced phenol quantified using its molar extinction coefficient (1310 M^−1^cm^−1^). Enzyme activities were expressed in U/L.

Serum concentrations of total cholesterol (TC), LDL-C, oxLDL, HDL-C, and TG were quantified employing commercially available enzyme-linked immunosorbent assay (ELISA) kits (Abcam, London, UK), strictly adhering to the manufacturer’s protocols. Measurements were performed using the varioskan LUX multimode microplate reader (Thermo Fisher Scientific, Waltham, MA, USA). The atherogenic lipid profile and the associated risk of coronary heart disease were subsequently determined by calculating the log(TG/HDL-C) ratio.

### 2.5. Statistical Analyses

Statistical analyses were performed using IBM SPSS version 25.0 software (IBM SPSS Inc., Chicago, IL, USA). Data normality was assessed using the Shapiro–Wilk test, skewness, kurtosis, histograms, and Q-Q plots. Normally distributed continuous variables were presented as mean ± standard deviation, while non-normally distributed variables were expressed as median [interquartile range]. Categorical variables were reported as frequency (n) and percentage (%). For the comparison of multiple independent groups with normal distribution, one-way ANOVA was used, followed by Tukey HSD for post hoc comparisons. The Kruskal–Wallis H test was applied for non-normally distributed data, with Bonferroni correction for post hoc comparisons. Given that the numerical variables followed a normal distribution, comparisons between the exercise groups were conducted using the unpaired Student’s *t*-test. Pearson Chi-squared test was used for categorical variables. Relationships between variables were evaluated using Spearman’s correlation analysis. Results were analysed with a 95% confidence interval, and *p*-values less than 0.05 were considered statistically significant.

## 3. Results

### 3.1. Demographic and Baseline Characteristics

Table 1 provides a comprehensive evaluation of the demographic and characteristic attributes of the exercise and sedentary groups, revealing no statistically significant differences among the groups in terms of age, cigarette smoking, alcohol intake, systolic blood pressure (SBP), and height with *p*-values of 0.357, 0.809, 0.291, 0.323, and 0.244, respectively. Additionally, the history and duration of exercise activity showed no significant differences among the exercise groups (*p* = 0.961 and *p* = 0.054, respectively). However, significant differences were noted in diastolic blood pressure (DBP), TC, LDL-C, HDL-C, TG, weight, BSA, BMI, waist-to-hip ratio (WHR), waist-to-height ratio (WHtR), abdominal skinfold thickness (AST), triceps skinfold thickness (TST), suprailiac skinfold thickness (SIST), skeletal muscle mass (SMM), body fat percentage (BFP), obesity degree (OD), FFMI, FMI, basal metabolic rate (BMR), frequency of physical activity, fitness score (FS), and MET-min/week (*p* < 0.001 for all comparisons). The VIBBE group exhibited significantly higher values in weight, BSA, BMI, and WHR compared to the NWS and MIAE groups (*p* < 0.001 for all), but these parameters did not differ significantly between the OOS and VIBBE groups (*p* > 0.05). Additionally, TC, LDL-C, weight, BSA, BMI, WHR, WHtR, TST, SIST, and OD levels were similar between the NWS and MIAE groups (*p* > 0.05). The VIBBE group had significantly lower DBP and TST levels compared to the NWS group (*p* < 0.05 for all), and higher values in weight, BSA, BMI, WHR, WHhR, SMM, OD, FFMI, and BMR (*p* < 0.05 for all). When compared to the OOS group, the VIBBE group displayed significantly lower DBP, LDL-C, TG, WHtR, AST, TST, SIST, BFP, OD, and FMI levels (*p* < 0.05 for all), and higher SMM, FFMI, and BMR levels (*p* < 0.05 for all). In contrast, the OOS group showed significantly higher TC, LDL-C, weight, BSA, BMI, WHR, WHtR, AST, TST, SIST, BFP, OD, and FMI levels compared to the MIAE group (*p* < 0.05 for all). These findings highlight the distinct physiological profiles associated with vigorous-intensity bodybuilding exercises and emphasise significant differences in body composition and metabolic parameters among the various exercise and sedentary groups.

### 3.2. Atherosclerosis Biomarkers

Table 2 presents a detailed comparison of atherosclerosis biomarkers across various exercise and sedentary groups. No significant differences in oxLDL levels were observed among the groups (*p* = 0.492). The VIBBE group showed no significant differences in PON1 and ARE activities, oxLDL levels, and log(TG/HDL-C) ratio compared to the NWS group (*p* > 0.05 for all). Similarly, the VIBBE group exhibited no significant differences in PON1 and ARE activities and oxLDL levels compared to the OOS group (*p* > 0.05 for all), but had a significantly lower log(TG/HDL-C) ratio (*p* < 0.05). The VIBBE group had significantly lower PON1 and ARE activities compared to the MIAE group (*p* < 0.05 for both). No significant differences were found between the OOS and NWS groups in PON1 and ARE activities, oxLDL levels, and log(TG/HDL-C) ratio (*p* > 0.05 for all). The MIAE group showed significantly higher PON1 and ARE activities compared to the NWS group (*p* < 0.05 for all), but no significant differences in oxLDL levels and log(TG/HDL-C) ratio (*p* > 0.05 for all). Compared to the OOS group, the MIAE group had significantly higher PON1 and ARE activities (*p* < 0.05 for all) and a significantly lower log(TG/HDL-C) ratio (*p* < 0.05). These findings underscore the distinct effects of VIBBE and MIAE on atherosclerosis biomarkers, highlighting the importance of tailored exercise regimens in managing cardiovascular risk.

### 3.3. Correlation Analysis

Table 3 presents a detailed evaluation of the correlations between demographic characteristics and atherosclerosis biomarkers among exercise and sedentary groups, revealing complex relationships that underscore the nuanced impacts of different exercise intensities and sedentary behaviours on cardiovascular health markers. Significant inverse correlations were observed between PON1 activity and BMI, WHR, WHtR, BFP, FMI, and OD (*rho* = −0.265, *p*= 0.003; *rho* = −0.213, *p*= 0.019; *rho* = −0.268, *p*= 0.003; *rho* = −0.188, *p*= 0.038; *rho* = −0.220, *p*= 0.015; *rho* = −0.260, *p*= 0.004, respectively). Similarly, ARE activity exhibited significant inverse relationships with BMI, WHtR, SMM, FFMI, OD, and MET-min/week (*rho* = −0.243, *p*= 0.007; *rho* = −0.187, *p*= 0.039; *rho* = −0.192, *p*= 0.034; *rho* = −0.233, *p*= 0.010; *rho* = −0.192, *p*= 0.034; *rho* = −0.298, *p*= 0.018, respectively). In contrast, oxLDL-C showed significant positive correlations with BMI, BSA, WHR, BFP, FMI, and OD (*rho* = 0.410, *p* < 0.001; *rho* = 0.407, *p* < 0.001; *rho* = 0.546, *p* < 0.001; *rho* = 0.454, *p* < 0.001; *rho* = 0.477, *p* < 0.001; *rho* = 0.475, *p* < 0.001, respectively). The log(TG/HDL-C) ratio also displayed significant positive correlations with BSA, WHR, WHtR, BFP, FMI, and OD (*rho* = 0.190, *p*= 0.036; *rho* = 0.281, *p*= 0.002; *rho* = 0.266, *p*= 0.003; *rho* = 0.355, *p* < 0.001; *rho* = 0.333, *p* < 0.001; *rho* = 0.217, *p*= 0.017, respectively), and a negative correlation with FS (*rho* = −0.243, *p*= 0.007). These findings highlight the intricate interplay between body composition, physical activity metrics, and atherosclerosis biomarkers, emphasising the significant role of tailored exercise regimens in managing cardiovascular risk.

## 4. Discussion

This study investigates the differential effects of vigorous-intensity bodybuilding exercises on atherosclerosis biomarkers, comparing them with moderate-intensity aerobic exercise, sedentary obese individuals, and normal-weight sedentary groups. This is the first study to examine these biomarkers in bodybuilders who do not use anabolic-androgenic steroids (AAS), filling a gap in the existing research. The focus is on oxLDL, log(TG/HDL-C), PON1, and ARE, which are pivotal in the pathogenesis and progression of atherosclerosis [7,8]. Elevated oxLDL and log(TG/HDL-C) levels contribute to oxidative stress and plaque formation [7], while PON1 and ARE have antioxidant properties that protect against atherosclerosis [21,37]. This aligns with findings by Martins et al. (2018) and Tyler and Thanos (2023), which indicate that resistance training positively impacts lipid profiles and metabolic health [38,39]. The findings aim to enhance understanding of how different exercise intensities and types affect cardiovascular health and guide exercise recommendations to reduce atherosclerotic risk.

Our results indicate that VIBBE significantly reduced LDL-C, TG, and the log(TG/HDL-C) ratio compared to OOS. However, these improvements were not statistically significant when compared to NWS, suggesting that the lipid-lowering effects may be primarily attributed to weight status rather than the exercise modality itself. Furthermore, VIBBE did not result in significant increases in HDL-C levels. This limited impact on enhancing the anti-atherogenic lipid profile underscores the need to evaluate the efficacy of VIBBE in cardiovascular risk reduction.

A pivotal finding is that VIBBE did not significantly enhance PON1 and ARE activities compared to sedentary groups (OOS and NWS). Considering the critical role of these enzymes in preventing oxidative modifications of LDL-C—a central event in atherogenesis [5,7,8,9]—this suggests that VIBBE may not effectively augment antioxidant defences. The lack of enhancement in PON1 and ARE activities in the VIBBE group may be due to the high-intensity nature of resistance training, which can increase the production of ROS [40]. This increased oxidative stress may overwhelm the body’s antioxidant capacity if not balanced with sufficient antioxidant intake or recovery time. The unchanged PON1 and ARE activities are significant, given their associations with increased oxidative stress and various systemic conditions, including neurodegenerative diseases, type 2 diabetes, NAFLD, CKD, and autoimmune disorders [22,23,24,25].

In contrast, participants engaged in MIAE demonstrated significant enhancements in PON1 and ARE activities, along with favorable lipid profiles characterised by increased HDL-C and reduced LDL-C and TG levels. This observation is consistent with extensive literature documenting the cardiovascular benefits of moderate-intensity aerobic exercise, which enhances antioxidant defences and reduces oxidative stress [18,21,41]. The elevated antioxidant enzyme activities in the MIAE group underscore the role of moderate aerobic activities in providing comprehensive cardiovascular protection [42]. Correlation analysis further supports this finding, revealing an inverse relationship between PON1 activity and measures of adiposity such as BMI, WHR, WHtR, BFP, FMI, and OD, suggesting that lower adiposity is associated with higher antioxidant enzyme activities.

Additionally, the OOS group exhibited higher levels of log(TG/HDL-C) and lower HDL-C compared to the MIAE group, highlighting the detrimental effects of obesity on cardiovascular health. These results align with established evidence that sedentary behaviour and obesity are significant risk factors for atherosclerosis and other cardiovascular diseases [26,43,44]. The elevated log(TG/HDL-C) and reduced HDL-C levels in sedentary individuals underscore the importance of regular physical activity in maintaining cardiovascular health [45]. Furthermore, the significant positive correlations between oxLDL-C or log(TG/HDL-C) and BMI, BSA, WHR, WHtR, BFP, FMI, and OD emphasise the critical need for physical activity to manage these risk factors effectively.

The differential effects of VIBBE and MIAE on atherosclerosis biomarkers may be attributed to their distinct physiological adaptations. Resistance training primarily induces muscle hypertrophy and increases strength, but may not sufficiently stimulate the antioxidant defence system [13,40]. The potential increase in ROS production during high-intensity resistance exercise could contribute to oxidative stress if antioxidant defences are not concurrently enhanced [40]. Conversely, moderate-intensity aerobic exercise induces favourable adaptations in oxidative metabolism, upregulates antioxidant enzymes, and improves endothelial function [16,20,45]. The increased PON1 and ARE activities observed in the MIAE group may result from enhanced gene expression and enzyme activation mediated by regular aerobic activity [21]. These enzymes play a crucial role in hydrolysing oxidised lipids, thereby preventing the formation of atherogenic oxLDL particles [8,9].

### 4.1. Limitations

Several limitations of this study must be acknowledged. First, the cross-sectional design precludes the establishment of causal relationships between exercise modalities and changes in atherosclerosis biomarkers, thereby limiting conclusions about the long-term effects of exercise. Second, reliance on self-reported data for dietary intake and lifestyle factors introduces potential biases that may compromise the accuracy of the findings. Third, the exclusion of key measurements, such as reactive oxygen species (ROS), apolipoprotein A1 (apoA1) levels, and HDL2/HDL3 subfractions, restricts our understanding of the underlying mechanisms of oxidative stress and lipid metabolism involved. Finally, the focus on male participants limits the generalizability of the results to females, particularly since estrogen is known to influence PON1 activity [46].

### 4.2. Future Directions

Subsequent study ought to include supplementary biomarkers, including C-reactive protein (CRP), homocysteine, and fibrinogen, to enhance the understanding of the inflammatory and thrombotic mechanisms implicated in atherosclerosis. Incorporating functional testing may yield information into cardiovascular performance and endothelial function. Furthermore, longitudinal studies that track individuals over time, either via direct follow-up or by utilising government medical databases, would be essential in evaluating the long-term impacts of intense bodybuilding and moderate exercise on cardiovascular health. Incorporating female respondents and persons from a wider age spectrum might improve the generalisability of the findings. Furthermore, examining the interplay between nutrition, supplements, and exercise will provide a more refined comprehension of how these factors collectively affect atherosclerotic biomarkers.

## 5. Conclusions

This study demonstrates that while vigorous-intensity bodybuilding exercises significantly reduce LDL-C, TG, and log(TG/HDL-C) levels compared to the OOS group, they do not yield significant improvements when compared to the NWS group, indicating a limited impact on lipid profile enhancement. Furthermore, no significant increases in the activities of key antioxidant enzymes, PON1 and ARE, were observed when compared to the sedentary groups. The limited effect of vigorous-intensity bodybuilding exercises on lipid profiles, coupled with the absence of a significant increase in antioxidant enzyme activities, suggests that this type of exercise may be insufficient to counteract oxidative stress, potentially increasing cardiovascular risk. In contrast, moderate-intensity aerobic exercise has been shown to enhance the activities of antioxidant enzymes such as PON1 and ARE, while promoting favorable lipid profiles, thereby reinforcing its protective role in cardiovascular health. Sedentary obese individuals exhibited higher log(TG/HDL-C) levels and lower HDL-C levels compared to the moderate-intensity exercise group, confirming the adverse effects of obesity and inactivity on atherosclerosis biomarkers. Overall, these findings indicate that different types of exercise exert distinct effects on atherosclerosis biomarkers in men, with moderate-intensity aerobic exercise offering superior cardiovascular protection.

## Figures and Tables

**Table 1 medicina-60-01717-t001:** Demographic, characteristics, and lipid profile of exercise and sedentary groups.

	VIBBE (*n* = 31)	NWS (*n* = 29)	OOS (*n* = 30)	MIAE (*n* = 32)	*F*, *H*, *t*, and *p* Values
Age (years) ^#^	28.55 ± 4.83	27.10 ± 5.98	28.67 ± 5.49	26.63 ± 5.52	*F* (3, 118) = 1.089, *p* = 0.357
Smoking	11	7	9	9	*p* = 0.809
Alcohol	3	5	2	7	*p* = 0.291
Systolic Blood Pressure (SBP, mmHg) ^#^	120.65 ± 16.76	123.64 ± 13.16	125.73 ± 15.28	118.65 ± 12.16	*F* (3, 118) = 1.448, *p* = 0.233
Diastolic Blood Pressure (DBP, mmHg) ^#^	66.74 ± 14.55 ^a,b^	76.25 ± 9.89	75.90 ± 12.70	69.19 ± 11.32	*F* (3, 117) = 4.575, *p* = 0.005
Lipid Profile					
Total Cholesterol (TC, mg/mL) ^α^	176 [45]	166 [64] ^d^	196 [58] ^f^	170 [25]	*H* (3) = 17.795, *p* < 0.001
Low Density Lipoprotein Cholesterol (LDL-C, mg/mL) ^α^	118 [44.2] ^b,c^	110 [45.5] ^d^	133 [42.7] ^f^	104 [26.4]	*H* (3) = 22.895, *p* < 0.001
High Density Lipoprotein Cholesterol (HDL-C, mg/dL) ^#^	41,1 ± 6.40	39.9 ± 5.85	37.9 ± 6.38 ^f^	42.5 ± 6.58	*F* (3, 118) = 2.880, *p* = 0.039
Triglyceride (TG, mg/dL) ^#^	110 ± 45.5 ^b^	124 ± 59.3	150 ± 53.3	119 ± 45.3	*F* (3, 118) = 3.458, *p* = 0.019
Body Composition and Anthropometry					
Height (cm) ^#^	176.45 ± 6.63	177.21 ± 5.28	178.00 ± 5.90	179.38 ± 5.68	*F* (3, 118) = 1.408, *p* = 0.244
Weight (kg) ^#^	87.62 ± 7.17 ^a,c^	71.34 ± 7.40 ^d^	91.22 ± 8.34 ^f^	72.62 ± 6.11	*F* (3, 118) = 59.322, *p* < 0.001
Body Surface Area (BSA, m^2^) ^#^	2.07 ± 0.12 ^a,c^	1.87 ± 0.12 ^d^	2.12 ± 0.21 ^f^	1.90 ± 0.10	*F* (3, 118) = 34.257, *p* < 0.001
Body Mass Index (BMI, kg/m^2^) ^#^	28.11 ± 0.99 ^a,c^	22.70 ± 1.97 ^d^	28.75 ± 1.64 ^f^	22.56 ± 1.43	*F* (3, 118) = 144.647, *p* < 0.001
Waist-to-Hip Ratio (WHR) ^#^	0.93 ± 0.03 ^a,c^	0.88 ± 0.04 ^d^	0.95 ± 0.02 ^f^	0.87 ± 0.03	*F* (3, 118) = 45.236, *p* < 0.001
Waist-to-Height ratio (WHtR) ^α^	0.51 [0.05] ^a,b,c^	0.47 [0.06] ^d^	0.57 [0.04] ^f^	0.45 [0.03]	*H* (3) = 86.265, *p* < 0.001
Abdominal Skinfold Thickness (AST, mm) ^#^	23.31 ± 8.84 ^b^	26.79 ± 9.07 ^d,e^	41.81 ± 5.41 ^f^	18.74 ± 4.93	*F* (3, 118) = 57.575, *p* < 0.001
Triceps Skinfold Thickness (TST, mm) ^α^	8.10 [5.60] ^a,b^	11.00 [7.50] ^d^	18.50 [15.13] ^f^	9.25 [4.00]	*H* (3) = 36.577, *p* < 0.001
Supra Iliac Skinfold Thickness (SIST, mm) ^α^	9.00 [15.00] ^b^	16.00 [10.75] ^d^	24.50 [11.00] ^f^	10.50 [12.38]	*H* (3) = 33.916, *p* < 0.001
Skeletal Muscle Mass (SMM, kg) ^α^	42.66 ± 4.54 ^a,b,c^	33.22 ± 2.72 ^d,e^	37.35 ± 4.18	36.53 ± 3.50	*F* (3, 118) = 31.928, *p* < 0.001
Body Fat Percentage (BFP, Kg) ^α^	15.40 [6.50] ^b^	17.20 [8.45] ^d,e^	28.85 [7.25] ^f^	12.10 [5.52]	*H* (3) = 73.887, *p* < 0.001
Obesity Degree (OD, %) ^α^	124.00 [14.00] ^a,c^	105.00 [16.00] ^d^	133.00 [17.50] ^f^	102.00 [10.50]	*H* (3) = 89.054, *p* < 0.001
Fat-Free Mass Index (FFMI) ^#^	23.96 ± 1.34 ^a,b,c^	18.66 ± 1.00 ^d,e^	20.51 ± 1.65	19.89 ± 1.33	*F* (3, 118) = 85.827, *p* < 0.001
Fat Mass Index (FMI) ^#^	4.14 ± 1.40 ^b,c^	4.04 ± 1.46 ^d,e^	8.23 ± 1.54 ^f^	2.67 ± 0.89	*F* (3, 118) = 97.759, *p* < 0.001
Basal Metabolic Rate (BMR, kCal/day) ^#^	1965 ± 162 ^a,b,c^	1633 ± 100 ^d,e^	1797 ± 155	1711 ± 121	*F* (3, 118) = 29.987, *p* < 0.001
Physical Activity					
History (year) ^#^	6.45 ± 5.08			5.93 ± 3.10	*t* (61) = 0.486, *p* = 0.961
Frequency (day/week) ^#^	4.81 ± 0.98			3.50 ± 0.95	*t* (61) = 5.371, *p* < 0.001
Duration (min) ^#^	81.29 ± 13.84			74.06 ± 15.21	*t* (61) = 1.971, *p* = 0.054
Fitness Score (FS) ^#^	92.90 ± 7.26			83.13 ± 5.50	*t* (61) = 6.037, *p* < 0.001
MET-min/week ^#^	2934 ± 850			1091 ± 293	*t* (61) = 11.581, *p* < 0.001

Parameters are expressed as ^#^: mean ± SD or ^α^: median [IQR]. Significant differences were identified as follows: ^a^: VIBBE vs. NWS, ^b^: VIBBE vs. OOS, ^c^: VIBBE vs. MIAE, ^d^: NWS vs. OOS, ^e^: NWS vs. MIAE, and ^f^: OOS vs. MIAE. Here, SD represents standard deviation, and IQR denotes interquartile range.

**Table 2 medicina-60-01717-t002:** Comparison of atherosclerosis biomarkers among VIBBE, NWS, OOS, and MIAE groups.

	VIBBE (*n* = 31)	NWS (*n* = 29)	OOS (*n* = 30)	MIAE (*n* = 32)	*F*, *H*, and *p* Values
Paraoxonase 1 (PON1) Activity(U/L) ^α^	90.0 [69.0] ^c^	64.0 [76.0] ^e^	81.0 [65.0] ^f^	116 [58.5]	H (3) = 12.019, *p* = 0.007
Arylesterase (ARE) Activity (U/L) ^α^	5803 [2045] ^c^	6101 [1253] ^e^	6495 [1700] ^f^	7082 [1688]	H (3) = 12.572, *p* = 0.006
Oxidised Low Density Lipoprotein (oxLDL, pg/mL) ^α^	1577 [440]	1488 [581]	1618 [224]	1528 [247]	H (3) = 2.411, *p* = 0.492
log(TG/HDL-C) ^#^	0.39 ± 0.20 ^b^	0.45 ± 0.24	0.57 ± 0.19 ^f^	0.42 ± 0.19	F (3, 118) = 4.351, *p* = 0.006

Parameters are expressed as ^#^: mean ± SD or ^α^: median [IQR]. Significant differences were identified as follows, ^b^: VIBBE vs. OOS, ^c^: VIBBE vs. MIAE, ^e^: NWS vs. MIAE, and ^f^: OOS vs. MIAE. Here, SD represents standard deviation, and IQR denotes interquartile range.

**Table 3 medicina-60-01717-t003:** Relationship between body composition and physical activity metrics with atherosclerosis biomarkers.

		BMI	BSA	WHR	WHtR	SMM	BFP	FFMI	FMI	OD	FS	MET-min/Week
PON1	*rho*	−0.265	−0.173	−0.213	−0.268	−0.122	−0.188	−0.173	−0.220	−0.260	−0.053	−0.230
	*p*	0.003	0.056	0.019	0.003	0.182	0.038	0.057	0.015	0.004	0.560	0.070
ARE	*rho*	−0.243	−0.160	−0.144	−0.187	−0.192	−0.020	−0.233	−0.076	−0.192	−0.146	−0.298
	*p*	0.007	0.079	0.114	0.039	0.034	0.824	0.010	0.404	0.034	0.110	0.018
oxLDL	*rho*	0.410	0.407	0.546	0.096	0.158	0.454	0.133	0.477	0.475	−0.137	0.106
	*p*	<0.001	<0.001	<0.001	0.291	0.081	<0.001	0.145	<0.001	<0.001	0.133	0.410
log(TG/HDL-C)	*rho*	0.133	0.190	0.281	0.266	0.007	0.355	−0.090	0.333	0.217	−0.243	−0.012
	*p*	0.144	0.036	0.002	0.003	0.943	<0.001	0.326	<0.001	0.017	0.007	0.925

PON1: Paraoxonase 1 activity, ARE: arylesterase activity (U/L), oxLDL: oxidised low-density lipoprotein, log(TG/HDL-C): logarithm of the triglyceride to high-density lipoprotein cholesterol ratio, BMI: body mass index, BSA: body surface area, WHR: waist-to-hip ratio, WHtR: waist-to-height ratio, SMM: skeletal muscle mass, BFP: body fat percentage, FFMI: fat-free mass index, FMI: fat mass index, OD: obesity degree, FS: fitness score, MET-min/week: metabolic equivalent of task minutes per week.

## Data Availability

Data are contained within the article.

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
