# Peer review of "Assessment of Paraoxonase 1 and Arylesterase Activities and Lipid Profile in Bodybuilders: A Comparative Study of Physical Activity and Anthropometry on Atherosclerosis"

_medicina, 2024, doi:10.3390/medicina60101717_

Round 1

Reviewer 1 Report

Comments and Suggestions for Authors

The manuscript by Celik et al entitled " Assessment of Paraoxonase 1 and Arylesterase Activities and Lipid Profile in Bodybuilders: A Comparative Study of Physical Activity and Anthropometry on Atherosclerosis". The text explores the relationship between physical activity, particularly bodybuilding, and cardiovascular health markers such as lipid profiles, Paraoxonase 1 (PON1), and Arylesterase (ARE) activities.

Strong points of the manuscript:

1.       The manuscript raises important topic such as lifestyle and the risk of civilization diseases

2.       The study design especially the groups are well divided (bodybuilders, individuals engaged in moderate exercise, overweight/obese sedentary individuals, and normal-weight sedentary individuals) strengthens the comparative analysis and helps highlight the diverse impacts of different lifestyle habits. Noteworthy the participants are very similar in age and height.

3.       Combining well-established markers such as lipid profile and elements of novelty such as oxidative stress markers

4.       The observations are interesting despite they are somehow surprising…

5.       Weakness of the study has been mentioned however not discussed or explained

Weak points:

1.       Data about PON1 and ARE requires to be updated. Over the past few years, significant advancements have been made in understanding the broader implications of PON1 and ARE especially in exercise etc..

2.       Abstract and Introduction: Authors might consider breaking up the text into smaller paragraphs for easier readability, especially when discussing different biomarkers or types of physical activity. The sentences are long and overflowing with information. For a general reader or non-experts, some sentences could be simplified,

3.       Results section: Please clarify where parameters are expressed as mean ± SD or median [IQR]

4.       Discussion:

It would be interesting if the received results were combined with CRP, homocysteine, or fibrinogen or even with some functional tests. Moreover, longitudinal observation of the participants by contact or official medical database monitoring.  Authors might suggest those in subsection “future direction”

It should be discussed that PON1 and ARE could also be involved in other conditions such as:

1. Neurodegenerative Diseases. Altered levels of these enzymes could indicate early signs of neurodegeneration.

2. Diabetes: Reduced PON1 activity has been noted in patients with type 2 diabetes.

3. Liver Diseases: Decreased PON1 activity has been linked to non-alcoholic fatty liver disease (NAFLD).

Comments on the Quality of English Language

 Abstract and Introduction: Authors might consider breaking up the text into smaller paragraphs for easier readability, especially when discussing different biomarkers or types of physical activity. The sentences are long and overflowing with information. For a general reader or non-experts, some sentences could be simplified.

There is some inconsistency in the tenses. For instance, in the discussion section, "The VIBBE group exhibited lower PON1 and ARE activities" would benefit from keeping the past tense consistent across the section.

Missing articles like "a" or "the" were noted in some sections

Reviewer 2 Report

Comments and Suggestions for Authors

In this study, Hakim Çelik et al. assessed the activities of Paraoxonase 1 (PON1) and Arylesterase Activities and Lipid Profile during vigorous and moderate-intensity aerobic exercise in male in Bodybuilders.

PON1 is well-known of its antioxidant and anti-atherogenic effects that prevents or reduces cardiovascular complications through different mechanisms by reducing plasma ox-LDL levels, reducing macrophage ability to uptake ox-LDL, reducing foam cell formation, promoting macrophage cholesterol efflux, and inhibiting MCP-1. 

1/ It has been reported that ROS may cause the oxidation of the PON1 protein via interaction with free sulfhydryl (SH) groups in Cysteine-284, which is considered the active site of PON1. Did the authors attempt to measure levels of ROS?

2/ Both protein levels and activities of PON1 are important. What were the levels of PON1 protein among different groups? PON1 activity normalized to protein levels?

3/ Did the authors collect data pre- and post-exercise absolute levels of ARE activity?

4/ Authors concluded and stated the following: “Vigorous-intensity bodybuilding improves lipid profiles by reducing LDL-C and TG levels but may decrease PON1 and ARE activities, potentially increasing oxidative stress”; however, the data from table 1 and table 2 are not fully in good agreement with the mentioned statement. LDL increased in group VIBBE vs. MIAE but no apparent significant effect on HDL nor TG levels (table 1). In contrast, PON1 & ARE activities were reduced in group VIBBE vs. MIAE (table 2). Additionally, LDL increased (not decreased) but TG deceased and no effect on HDL in VIBBE vs. OOS (table 2). These findings do not support your conclusion. Please explain discrepancies.

5/ Most of the PON1 activity carried by HDL-C is found in the HDL3-C small sub-fraction of HDL. Also, PON1 concentrations were significantly and positively associated with HDL-C, HDL3-C and Apo A1. Did the authors measure levels of apoA1 and HDL2/HDL3 among the groups?

6/ Authors have mentioned that the study was conducted only in males and considered this as a limitation of study. Another reason to exclude female is because estrogen level was previously shown to have a stimulating effect on the PON1 enzyme which could be a confounding factor in this specific study.

7/ Authors may quote and discuss relevant data from recent study (Otocka-Kmiecik, A.; Orłowska-Majdak, M.; Stawski, R.; Szkudlarek, U.; Padula, G.; Gałczyński, S.; Nowak, D. Effect of Exercise Repetitions on Arylesterase Activity of PON1 in Plasma of Average-Trained Men—The Dissociation between Activity and Concentration. Antioxidants 2023, 12, 1296. https://doi.org/10.3390/antiox12061296).

Comments on the Quality of English Language

No major issues with English language, however proofreading is encouraged.

Round 2

Reviewer 1 Report

Comments and Suggestions for Authors

I have reviewed the authors' revisions and their responses to my comments. I am writing to inform you that this revised version is worthy of publication. The authors have adequately addressed the concerns raised, and the manuscript is now suitable for inclusion in the journal.

Thank you for the opportunity to contribute to this process.

Reviewer 2 Report

Comments and Suggestions for Authors

Authors have successfully addressed all the concerns raised during the first round of review. Manuscript is very much improved.

Minor: A reference regarding modulation of PON1 by estrogen should be given for the last sentence within limitations section.

For example: 

Eraldemir, F.C.; Korak, T. Paraoxonases, Oxidative Stress, and Breast Cancer. In Cancer Oxidative Stress Diet. Antioxidants; Academic Press: Cambridge, MA, USA, 2021; pp. 3–14.

Ahmad S, Scott JE. Estradiol enhances cell-associated paraoxonase 1 (PON1) activity in vitro without altering PON1 expression. Biochem Biophys Res Commun. 2010 Jul 2;397(3):441-6.